# Meta-Analysis of In-Vitro Bonding of Glass-Ionomer Restorative Materials to Primary Teeth

**DOI:** 10.3390/ma14143915

**Published:** 2021-07-14

**Authors:** Tamara Peric, Evgenija Markovic, Dejan Markovic, Bojan Petrovic

**Affiliations:** 1School of Dental Medicine, Clinic for Pediatric and Preventive Dentistry, University of Belgrade, Dr Subotića 11, 11000 Belgrade, Serbia; dejan.markovic@stomf.bg.ac.rs; 2School of Dental Medicine, Clinic of Orthodontics, University of Belgrade, 11000 Belgrade, Serbia; evgenija.markovic@stomf.bg.ac.rs; 3Faculty of Medicine, Dentistry Clinic of Vojvodina, Department of Pediatric and Preventive Dentistry, University of Novi Sad, 21000 Novi Sad, Serbia; BOJAN.PETROVIC@mf.uns.ac.rs

**Keywords:** primary teeth, restoration, glass-ionomer cements, bond strength test, meta-analysis

## Abstract

Restoration of primary teeth is among the main clinical applications of glass-ionomer cements (GIC). The aim of the study was to review and summarize existing evidence of *in vitro* bond strength of glass-ionomer (GI) restoratives to enamel and dentin of primary teeth. A literature search was performed in PubMed/Medline, Scopus, Web of Science, Cochrane, and Google Scholar databases to identify studies published until April 2021. The search strategy was: (“glass”) and (“ionomer”) and (“primary” or “deciduous”) and (“bond” or “tensile” or “shear”). Two researchers independently retrieved articles that reported on the bond strength of GIC to primary dentin and/or enamel. The meta-analysis was performed to compare the bond strength values of conventional (C) GIC and resin-modified (RM) GIC to different substrates. From 831 potentially eligible articles, 30 were selected for the full-text examination, and 7 were included in the analysis. Studies were rated at high (3), medium (3), and low (1) risk of bias. RM-GIC showed higher bond strength to primary enamel and dentin compared to the C-GIC. Meta-analysis of *in vitro* studies, evaluating bonding properties of GI restoratives to primary teeth, suggests the superior performance of RM-GIC. However, there is a lack of studies that examine the properties of novel GI formulations.

## 1. Introduction

Glass-ionomer cements (GIC) have been used in dentistry for almost five decades. With many favorable properties such as chemical bonding to the tooth structure, potential cariostatic effect due to fluoride release, biocompatibility, acceptable aesthetics, resistance to microleakage, and its dimensional stability, the applicability of GIC shows great potential [1,2]. In the context of minimally invasive dentistry, as well as restorative pediatric dentistry, GIC are a very popular choice for direct restorations due to a relatively short and simple application procedure and reduced occurrence of secondary caries [3]. The main drawbacks of conventional GIC are brittleness and relatively low wear resistance [4]. Resin composite materials possess favorable aesthetics and physical properties, as well as superior bonding performance in a controlled environment, but their application is often demanding, time-consuming, and more sensitive compared to the use of GIC. In pediatric dentistry, when the use of a rubber dam is challenging and patient cooperation is limited, GIC can be used to successfully restore both primary and permanent teeth [5].

Restoration of primary teeth is among the main clinical applications of glass-ionomer (GI) restoratives. Although all types of GIC can be used to restore primary teeth, conventional (C) GIC for restoration of multisurface cavities show higher failure rates [6], while resin-modified (RM) GIC demonstrate clinical success comparable to resin composites (CR) in primary dentition [6,7]. The introduction of newer generations of GIC has provided further opportunities for a wider variety of applications in permanent dentition. Conversely, there is still not enough information on the performances of novel GI formulations as direct restorative materials in primary teeth.

A large segment of the dental materials-related literature is based on *in vitro* studies [8], while clinical reports represent less than 10% of the total research activity [9]. Even though it is not possible to completely simulate the complex biological aspects of the oral environment, laboratory models are of great importance for dental research. Despite wide variations in testing protocols, bond strength measuring might predict the clinical effectiveness of dental restoratives [10]. When it comes to GI restoratives adhesion to primary teeth, there are publications evaluating bond strength of GI and CR materials in primary teeth, but little comparative work between various types of GIC has been conducted. The aim of the study was to review and summarize existing evidence of *in vitro* bond strength of GI restoratives to enamel and dentin of primary teeth.

## 2. Materials and Methods

This systematic review was prepared in accordance with the Preferred Reporting Items for Systematic Reviews and Meta-analyses (PRISMA 2020) Statement [11].

In order to identify as many eligible studies as possible, the search process was carried out in a thorough, objective, and reproducible way. To critically review the relevant *in vitro* studies, which reported on bonding properties of restorative GIC in primary teeth, a literature search was performed in PubMed/Medline, Scopus, Web of Science, Cochrane, and Google Scholar databases to identify studies published until April 2021. The search strategy was: (“glass”) and (“ionomer”) and (“primary” or “deciduous”) and (“bond” or “tensile” or “shear”).

Of obtained papers, only those published in English language were considered. Two researchers independently retrieved articles that reported results of *in vitro* bond strength of GIC to primary dentin and/or enamel for further assessment. Researchers were not blinded to article authors, institutions, and journal names. They examined titles and abstracts, removed duplications, reported exclusion reasons, retrieved, and examined the full text of potentially relevant articles in order to extract data. Disagreements were resolved by discussion. Initially, studies were excluded if they were identified as the following: (1) not *in vitro* studies, (2) did not evaluate bond strength, (3) the substrate was other than human primary enamel and/or dentin, and (4) studies that evaluated bond strength of GI orthodontic materials, luting materials, or experimental materials. The search was complemented with references that were hand-searched in papers selected for the analysis.

The following data were extracted from included studies: publication details (authors, title, year of publication), characteristics of the material (type of glass-ionomer- conventional or resin-modified, form of the material- capsulated or powder/liquid), sample (sample size, type of tooth- anterior or molars, tooth surface- buccal/lingual, mesial/distal, or occlusal, type of surface- flat or cavity, substrate- enamel or dentin, preconditioning of the substrate, adhesive area, storage- time and medium), type and outcome (mean bond strength) of the test, and failure mode. If needed, authors were contacted via e-mail for clarification regarding missing or unclear information. When the same data were reported in different publications, only one paper was considered.

The risk of bias was based on and adapted from a previous study [12]. To establish the quality of the study, the following parameters were assessed: random assignment of the teeth to the experimental groups, description of sample size calculation, the same number of teeth/specimens per group, restorative materials applied according to the manufacturers’ instructions, materials and testing procedures performed by a single operator, specimens tested by a blinded operator, and failure mode evaluation. If the parameter was described, the paper was marked with a ‘yes’, otherwise it received a ‘no’. Based on the ratings of existing items, the risk of bias was expressed as low (1–3 items), moderate (4–5 items), and high (6–7 items).

The statistical analysis started with a qualitative description of included studies and presentation of their results. The pooled effect estimates were calculated through a random-effects analysis by comparing the mean differences between the bond strength values of C-GIC and RM-GIC to primary enamel or dentin. The meta-analysis was performed to compare C-GIC and RM-GIC when applied in different substrates, such as enamel or dentin. Immediate and long-term bond strengths of GIC were analyzed separately. The presence of heterogeneity was analyzed via inconsistency (I²) with 95% CI. The I^2^ results were interpreted as follows: 0–40% = might not be important; 30–60% = may represent moderate heterogeneity; 50–90% = may represent substantial heterogeneity; 75–100% = considerable heterogeneity [13]. The analyses were performed using the Review Manager (RevMan) Version 5.4.1. (The Cochrane Collaboration, 2020, London, United Kingdom). The statistical significance was set at *p* ≤ 0.05 (*Z* test).

## 3. Results

The literature search revealed a total of 831 articles. After removing the duplicates, 541 articles were retrieved. Of these, 30 were selected for the full-text reading. One study was selected for reading through hand search of the references, but it did not meet the inclusion criteria. A total of seven studies that tested both C-GIC and RM-GIC in primary teeth were included in the meta-analysis. A flow chart of the study selection process and the reasons for exclusions are shown in Figure 1.

A detailed description of the included studies is provided in Table 1. The selected studies were published between 2002 and 2018. Bonding to sound dentin was evaluated in most of the studies [14,15,16,17,18]. One study investigated both adhesion to sound and caries affected dentin (CAD) [15]. Two studies evaluated adhesion to both enamel and dentin [19,20]. Sample size ranged from 6 to 32 teeth per group. However, for one study that employed microtensile bond strength [15], the final number of specimens (sticks) remained unclear.

The most frequently tested materials were Fuji IX (GC Int., Tokyo, Japan) and Vitremer (3M ESPE, Seefeld, Germany) in the liquid/powder form. In selected studies, materials were prepared and applied as per the manufacturer’s instructions. Dental tissues were conditioned in the following manner: (1) 20% polyacrilyc acid for 10 s [14,16], (2) 10% polyacrilyc acid for 20 s [17,18], (3) 10% polyacrilyc acid for 10 s [19,20], (4) 25% polyacrilyc acid for 10 s [15], or (5) Vitrebond primer—a mixture consisting of polycarboxylic acid and 2-hydroxyethylmethacrylate (HEMA) [15,19,20]. However, only 3 studies [15,19,20] reported the application of surface protection glaze/gloss, after the setting of GIC.

Results of the microtensile bond strength test were presented in 2 studies [14,15], microshear bond strength test was employed in 2 studies [19,20], shear bond strength test was used in 2 studies [16,18], and the results of the tensile bond strength test were reported in 1 study [17].

Four studies [14,16,17,18] evaluated immediate bond strength, while 3 studies reported on both short and long-term bond strength [15,19,20]. In the majority of studies, short-term storage for 24 h was evaluated, except for the study performed by Somani et al. (7–10 days) [18]. Rekha et al. [17] did not report exact storage time. Duration of long-term storage varied between 12 [19,20] and 24 months [15].

The risk of bias was high in 3 (43%) studies, 3 (43%) studies had moderate risk, and one study (14%) had low risk of bias (Table 2).

Statistical pooling of bond strength (CI 95%) is presented in Table 3. Overall, RM-GIC showed higher bond strength compared to C-GIC (Figure 2 and Figure 3). All authors reported significantly better performance of RM-GIC compared to C-GIC, except for Pacifici et al. [16], who did not record a significant difference between C-GIC and RM-GIC. No significant differences in bond strength values were reported between sound dentin and CAD. None of the studies included in the present analysis compared bond strength of GIC to sound enamel and dentin. Analysis of bond strength to sound dentin revealed higher values when microtensile tests were used. The predominance of adhesive/mixed failure mode was observed in 80% of analyzed studies.

## 4. Discussion

Dental caries is one of the main public health problems in many countries [21], with early childhood caries being a global challenge [22]. Modern caries management includes various approaches, such as: no caries removal techniques (non-restorative caries control, caries arresting methods, and sealing techniques), and operative interventions, which are recommended to be minimally invasive [7]. When it comes to the operative treatment, choosing the ideal restorative material for pediatric patients is still challenging. As the use of dental amalgam has been discontinued in many countries, the focus has been shifted to CR and GIC.

There is a general agreement that preformed metal crowns are the best choice for restoration of primary teeth with extensive lesions [7], but there is still inconsistency regarding the choice of the best conventional restorative material [5,23]. Several review articles favored RM-GIC, CR, and compomers rather than C-GIC and metal-reinforced GIC, due to their higher failure rates [6,24,25]. However, those reviews included studies that investigated dated materials, which are either rarely used or no longer available. Another review [5] included recent studies that examined behavior of new generations of restorative materials. No significant differences in retention, marginal discoloration, marginal adaptation, anatomical form, and wear between GIC (both C-GIC and RM-GIC) and CR for restoration of proximal cavities in primary teeth were found, except for the occurrence of secondary caries. It was emphasized that GIC had better ability to prevent secondary caries compared to the other restorative materials [5,26,27], especially in occluso-proximal restorations [27].

There are clear differences between laboratory and clinical findings, primarily because diverse influencing factors of the complex oral environment are difficult to simulate in laboratory conditions [9]. However, *in vitro* testing enables examination of a single parameter, and provides valuable initial information regarding the physico-mechanical characteristics of dental material.

Several key parameters are responsible for the success and longevity of dental restorations. However, a strong bond between the restorative material and the tooth substrate is crucial, not only from a mechanical, but also from a biological and aesthetic perspective [28]. GIC are characterized by chemical bonding to enamel and dentin, which is attained through interactions between hydroxyapatite and polycarboxylate radicals [29]. The present analysis showed better bonding properties of RM-GIC to primary enamel and dentin compared to C-GIC, both immediately and after long-term storage. RM-GIC offer both micromechanical interlocking and chemical bonding to tooth substrate [30] and provide higher bond strength values and better long-term stability [15]. RM-GIC consist of two components: a conventional GI and a resin. Both components affect mechanical properties, setting mechanism, and adhesive strength. RM-GIC bond to tooth structure by two mechanisms: chemically, through ionic bonding of the carboxyl group to the calcium ions of the tooth substrate (typical for GI component), and mechanically, by interlocking of the resinous component and conditioned tooth surface (polyacrylic conditioning provides slightly more retentive surface than not conditioned tooth structures) [31]. HEMA, as a component of RM-GIC, enhances monomer diffusion into the demineralized dentin matrix, and entanglement with its components. The addition of HEMA facilitates the formation of “hybrid” layers. The “hybrid” layer is the formation of a transitional zone of resin-reinforced dentin, essential for achieving high bond strength [32]. Like the abovementioned clinical reviews [6,24,25], the present study included articles that evaluated older GIC formulations. Different modifications were introduced in order to overcome the disadvantages of C-GIC. To the best of our knowledge, there are no reports on bonding properties of newer GI materials, such as novel high-viscosity C-GIC formulations in capsulated form and recently developed glass-hybrid materials to primary teeth.

In clinical practice, restorative materials bond to various substrates. Minimum intervention caries removal principles comprise preservation of remineralizable tooth tissues [33] such as CAD. It is a known fact that CR micromechanical bond strength to CAD is usually lower than to sound dentin [34], but it seems that partial dentin demineralization does not affect GIC chemical bonding. After the removal of infected dentin, placing GIC over CAD will seal the lesion and help healing of pulpo-dentinal complex and remineralization of underlying affected tissue [35]. Although anecdotally reported, published papers regarding interactions between GIC and affected primary dental tissues are rare. Calvo et al. [15] found no differences in short and long-term microtensile bond strength of C-GIC, RM-GIC, and nano-ionomer bonded to either sound dentin or CAD. On the contrary, Çehreli et al. [28] reported significantly lower microtensile bond strength of various resin-based materials, including RM-GIC, bonded to primary CAD, after 18-month of storage. However, the two studies used different methods for the preparation of artificial caries lesions, and, therefore, results cannot be compared.

The present meta-analysis included three papers that evaluated long-term bond strength. Tedesco et al. [19,20] reported stability of the adhesive interface of both C-GIC and RM-GIC and primary enamel and dentin during 12-month water storage. On the other hand, Calvo et al. [15] reported 24-month bond strength stability for RM-GIC only. Immediate and long-term bonding was tested extensively, but it has been emphasized that aged bond strength *in vitro* correlates better with clinical retention rates, and is of greater importance in predicting the clinical efficacy of dental restoratives [10].

Authors of the studies investigating the performance of GIC *in vitro* encountered similar problems with either interpretation of results or comparison with different types of materials tested in the same way [36]. The adhesion between GIC and dental tissues is very delicate, and technique sensitive [37]. The biggest inference errors can be made when the bond strength of restorative materials with different sensitivity to tests and preparation techniques are uncritically compared. In the present study, substantial heterogeneity was observed among studies that evaluated the immediate bond strength. Numerous factors such as: discrepancies in the sample size, properties of tested materials, storage conditions, testing methods, etc., could have influenced the heterogeneity. Interestingly, the highest heterogeneity was observed between studies completed by the same authors, in similar experimental conditions, regarding the immediate bond strength of GIC to enamel. When the long-term bond strength to enamel was evaluated by the same authors, the heterogeneity was moderate, and non-important heterogeneity was found between studies regarding the long-term bond strength to dentin.

Various bond strength testing techniques were used to assess the GIC bonding to primary teeth *in vitro*. The strongest bond of GI restoratives was reported when the microtensile bond strength test was used. The test uses a small cross-section area, and the stress distribution is homogenous [10]. The number of defects that might occur at the bonding interface is minimal, and initiation of cracks that lead to bond failure is reduced [14], which is in accordance with the findings of present analysis. Conversely, microshear bond strength was lower in comparison to shear bond strength. The quality of bond between tooth and GIC depends on interphase integrity, tooth conditioner, storage conditions, and performed tests [38]. A typical finding for GIC is cohesive failure in the material [14,38]. However, the predominance of adhesive/mixed failures was reported in 80% of studies included in the present review. The possible explanation might be related to the specifics of the testing procedures (shear and microshear test), or primary tooth-GIC interfacial strength might be weaker than in the bulk of GIC.

The authors acknowledge a few limitations of the present study. We examined *in vitro* performance of GI restoratives during a long period of time, in which formulation, testing protocols, and imaging have been changing. That is likely one of the reasons of the high risk of bias and heterogeneity of the included studies. Although novel formulations of GIC have been introduced, studies on their performance are lacking.

## 5. Conclusions

The meta-analysis of *in vitro* studies evaluating bonding properties of GI restorative materials to primary teeth suggests superior adhesion properties of RM-GIC to primary enamel and dentin, both immediately and after long-term storage. Finding an ideal restorative material for primary teeth is an ongoing quest, which could be sped up with further investigations of novel GIC. Development and standardization of test protocols, as well as an extension of the storage period, are pertinent in obtaining significant results. Continuous improvement of dental materials holds great promise for finding the ideal GI restorative for primary teeth.

## Figures and Tables

**Figure 1 materials-14-03915-f001:**
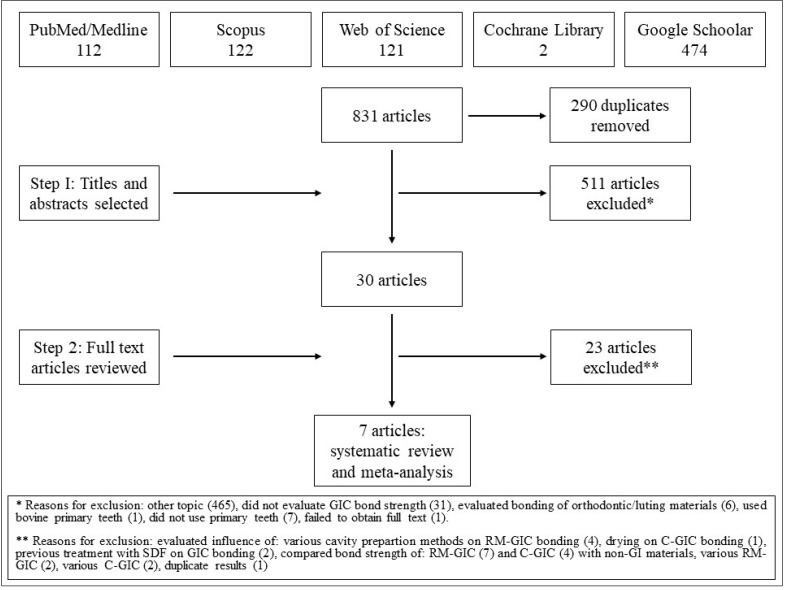
Flow chart of study selection.

**Figure 2 materials-14-03915-f002:**
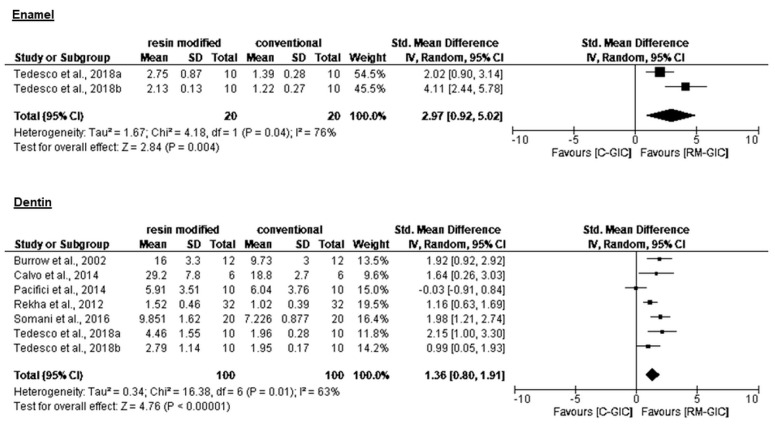
Forest plot for the meta-analysis of C-GIC and RM-GIC immediate bond strength in primary enamel and dentin.

**Figure 3 materials-14-03915-f003:**
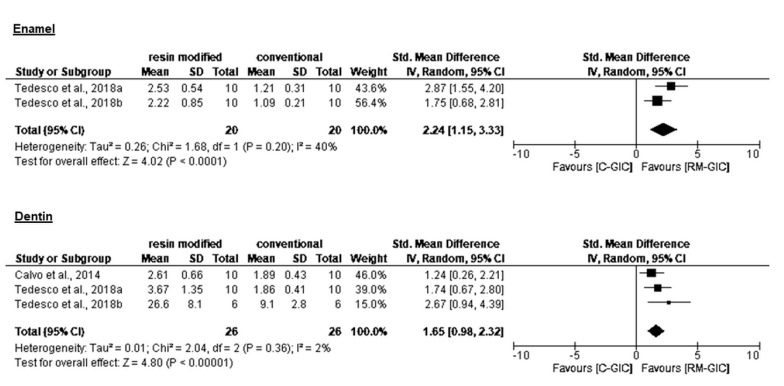
Forest plot for the meta-analysis of C-GIC and RM-GIC long-term bond strength in primary enamel and dentin.

**Table 1 materials-14-03915-t001:** Descriptive data of the included studies.

Study	Country	GIC Type	Commercial Brand	GIC Formulation	Tooth Type	Tooth Surface	Type of Surface	Substrate	Number of Teeth/Specimens (per Group)	Type of Test	Adhesive Area (mm²)	Storage	Bond Strength [MPa] (Mean (SD))
Burrow et al., 2002 [14]	Australia	conventional	Fuji IX	capsulated	molar	occlusal	flat	dentin	6/12	microtensile	11.3	24 h in tap water	9.7 (3.0)
		resin modified	Fuji II LC	capsulated	molar	occlusal	flat	dentin	6/12	microtensile	11.3	24 h in tap water	16.0 (3.3)
Calvo et al., 2014 [15]	Brazil	conventional	Ketac Molar	powder/ liquid	molar	occlusal	cavity	dentin	6/9–12	microtensile	0.8	24 h in distilled water	18.81 (2.65)
		resin modified	Vitremer	powder/ liquid	molar	occlusal	cavity	dentin	6/9–12	microtensile	0.8	24 h in distilled water	29.24 (7.84)
		conventional	Ketac Molar	powder/ liquid	molar	occlusal	cavity	caries affected dentin	6/9–12	microtensile	0.8	24 h in distilled water	14.52 (0.78)
		resin modified	Vitremer	powder/ liquid	molar	occlusal	cavity	caries affected dentin	6/9–12	microtensile	0.8	24 h in distilled water	24.9 (5.74)
		conventional	Ketac Molar	powder/ liquid	molar	occlusal	cavity	dentin	6/9–12	microtensile	0.8	2 y in distilled water	9.1 (2.8)
		resin modified	Vitremer	powder/ liquid	molar	occlusal	cavity	dentin	6/9–12	microtensile	0.8	2 y in distilled water	26.6 (8.1)
		conventional	Ketac Molar	powder/ liquid	molar	occlusal	cavity	caries affected dentin	6/9–12	microtensile	0.8	2 y in distilled water	9.7 (1.3)
		resin modified	Vitremer	powder/ liquid	molar	occlusal	cavity	caries affected dentin	6/9–12	microtensile	0.8	2 y in distilled water	19.1 (2.2)
Pacifici et al., 2013 [16]	Italy	conventional	Fuji IX	capsule	molar	occlusal	flat	dentin	10/-	shear	7.1	24 h in 100% humidity	6.04 (3.76)
		resin modified	Fuji II LC	capsule	molar	occlusal	flat	dentin	10/-	shear	7.1	24 h in 100% humidity	5.91 (3.51)
Rekha et al., 2012 [17]	India	conventional	Fuji IX	not reported	molar	occlusal	flat	dentin	32/-	tensile	12.6	Ringer’s solution	1.02 (0.39)
		resin modified	Fuji II LC	not reported	molar	occlusal	flat	dentin	32/-	tensile	12.6	Ringer’s solution	1.52 (0.46)
Somani et al., 2016 [18]	India	conventional	Fuji IX	not reported	molar	buccal/ lingual	flat	dentin	20/-	shear		7–10 d in distilled water thermo-cycling 500×	7.23 (0.88)
		resin modified	Fuji II LC	not reported	molar	buccal/ lingual	flat	dentin	20/-	shear		7–10 d in distilled water thermo-cycling 500×	9.85 (1.62)
Tedesco et al., 2018a [19]	Brazil	conventional	Fuji IX	powder/ liquid	molar	buccal/ lingual	flat	enamel	10	microshear	0.45	24 h in distilled water	1.39 (0.28)
		resin modified	Vitremer	powder/ liquid	molar	buccal/ lingual	flat	enamel	10	microshear	0.45	24 h in distilled water	2.75 (0.87)
		conventional	Fuji IX	powder/ liquid	molar	occlusal	flat	dentin	10	microshear	0.45	24 h in distilled water	1.96 (0.28)
		resin modified	Vitremer	powder/ liquid	molar	occlusal	flat	dentin	10	microshear	0.45	24 h in distilled water	4.46 (1.55)
		conventional	Fuji IX	powder/ liquid	molar	buccal/ lingual	flat	enamel	10	microshear	0.45	12 mo in distilled water	1.21 (0.31)
		resin modified	Vitremer	powder/ liquid	molar	buccal/ lingual	flat	enamel	10	microshear	0.45	12 mo in distilled water	2.53 (0.54)
		conventional	Fuji IX	powder/ liquid	molar	occlusal	flat	dentin	10	microshear	0.45	12 mo in distilled water	1.86 (0.41)
		resin modified	Vitremer	powder/ liquid	molar	occlusal	flat	dentin	10	microshear	0.45	12 mo in distilled water	3.67 (1.35)
Tedesco et al., 2018b [20]	Brazil	conventional	Fuji IX	powder/ liquid	molar	buccal/ lingual	flat	enamel	10	microshear	0.45	24 h in distilled water 7 d in saline	1.22 (0.27)
		resin modified	Vitremer	powder/ liquid	molar	buccal/ lingual	flat	enamel	10	microshear	0.45	24 h in distilled water 7 d in saline	2.13 (0.13)
		conventional	Fuji IX	powder/ liquid	molar	occlusal	flat	dentin	10	microshear	0.45	24 h in distilled water 7 d in saline	1.95 (0.17)
		resin modified	Vitremer	powder/ liquid	molar	occlusal	flat	dentin	10	microshear	0.45	24 h in distilled water 7 d in saline	2.79 (1.14)
		conventional	Fuji IX	powder/ liquid	molar	buccal/ lingual	flat	enamel	10	microshear	0.45	12 mo in distilled water	1.09 (0.21)
		resin modified	Vitremer	powder/ liquid	molar	buccal/ lingual	flat	enamel	10	microshear	0.45	12 mo in distilled water	2.22 (0.85)
		conventional	Fuji IX	powder/ liquid	molar	occlusal	flat	dentin	10	microshear	0.45	12 mo in distilled water	1.89 (0.43)
		resin modified	Vitremer	powder/ liquid	molar	occlusal	flat	dentin	10	microshear	0.45	12 mo in distilled water	2.61 (0.66)

**Table 2 materials-14-03915-t002:** Risk of bias.

Study	Random Allocation	Sample Size Calculation	Same Sample Size per Group	Manufacturer’s Instructions Followed	Single Operator	Blinded Operator	Failure Mode Evaluation	Risk of Bias
Burrow et al. [14], 2002	Yes	No	Yes	Yes	Yes	No	Yes	Medium
Calvo et al. [15], 2014	Yes	No	Unclear	Yes	No	No	Yes	High
Pacifici et al. [16], 2013	Yes	No	Yes	Yes	No	No	Yes	Medium
Rekha et al. [17], 2012	Yes	No	Yes	Yes	No	No	No	High
Somani et al. [18], 2016	No	No	Yes	Yes	No	No	No	High
Tedesco et al. [19], 2018a	Yes	No	Yes	Yes	Yes	Yes	Yes	Low
Tedesco et al. [20], 2018b	Yes	No	Yes	Yes	Yes	No	Yes	Medium

**Table 3 materials-14-03915-t003:** Pooling of immediate bond strength data (MPa) considering the bond strength test in various substrates in primary teeth.

Substrate	Overall	Microtensile	Tensile	Microshear	Shear
Sound enamel	2.97 (0.92–5.02) *I*^2^ = 76% *n* = 2	-	-	2.97 (0.92–5.02) *n* = 2	-
Sound dentin	1.36 (0.80–1.91) *I*^2^ = 63% *n* = 7	1.83 (1.02–2.64) *n* = 2	1.16 (0.63–1.69) *n* = 1	1.52 (0.38–2.66) *n* = 2	0.98 (−0.99–2.95) *n* = 2
Caries-affected dentin	2.36 (0.74–3.97) *n* = 1	2.36 (0.74–3.97) *n* = 1	-	-	-

## Data Availability

Data sharing is not applicable to this article.

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
