# Peer review of "Meta-Analysis of In-Vitro Bonding of Glass-Ionomer Restorative Materials to Primary Teeth"

_materials, 2021, doi:10.3390/ma14143915_

Round 1
Reviewer 1 Report
The manuscript materials-1263606 Meta-Analysis of In-Vitro Bonding of Glass-Ionomer Restorative Materials to Primary Teeth; is a review article on the application of glass-ionomer cement (GIC) for primary teeth restoration. Such review is vital for the field of dental materials used for restoration processes. In contrast, restorative dental materials are constantly developed, and there is a need to investigate the performances of modern glass-ionomer cement. The study's main aim was to review and summarize the present evidence of in vitro bond strength of GI restoratives to enamel and dentin of primary teeth. The Authors realized a comprehensive literature search in PubMed/Medline, Scopus, Web of Science, Cochrane and Google Scholar databases to identify studies on in vitro bonding of GIC (articles published until April 2021). The searching strategy was sound, and research risks were also shown. Realized analysis of in vitro studies evaluating bonding properties of GI restoratives to primary teeth suggests its good performance, especially for RM-GIC. Such analysis was performed to compare C-GIC and RM-GIC when applied in different substrates, such as enamel or dentin. The data were collected from publication details, characteristics of the material and samples, mean bond strength and failure mode. The article has shown that by measuring bond strength, the effectiveness of dental restoratives might be predicted. It has good scientific value to systematizing the methods and guides their advantages in tailoring the GIC. I appreciate showing these research results, but I think it would also be worth offering some images of in vitro results, mechanism of restorations with a different type of GIC. Summing up, the presented review is essential for dentistry. The compilation of the results obtained for GIC materials in perspective will give the Authors and other researchers possibilities to produce the novel restorative materials. The manuscript has good scientific value but should be upgraded to attract the Readers' attention in such a form. As I read the manuscript, I also found some mandatory errors that the authors should correct before the publication. Some detailed suggestions are listed in the following points:
- The conclusions in the article are too general. I suggest extending and re-editing the conclusions. I think The Authors should also highlight the main outlook of GIC applications here.
- Figures 2,3: The pictures should be upgraded. In such form, it has low quality of presentation and scientific value.
- Sections should be more precisely named
Materials and Methods better to say: (Literature) Research Methodology
Results better to say (Literature) Research Results
Discussion of the Results - Materials and Methods section:
Line 4 – invitro / should be – in vitro
There are no more comments that I felt to comment on. In general, the article requires editorial refinement and proofreading. The issues presented in the article are important for researchers of primary teeth restoration processes and suitable for publication as a review article in the Journal Materials, but to increase their scientific value manuscript should be again analyzed by the Authors. I also want to point out that the Authors should format the manuscript according to the journal's and the MDPI publishing house guidelines. I recommend the paper for publication in Journal Materials after major revision.
Author Response
The manuscript materials-1263606 Meta-Analysis of In-Vitro Bonding of Glass-Ionomer Restorative Materials to Primary Teeth; is a review article on the application of glass-ionomer cement (GIC) for primary teeth restoration. Such review is vital for the field of dental materials used for restoration processes. In contrast, restorative dental materials are constantly developed, and there is a need to investigate the performances of modern glass-ionomer cement. The study's main aim was to review and summarize the present evidence of in vitro bond strength of GI restoratives to enamel and dentin of primary teeth. The Authors realized a comprehensive literature search in PubMed/Medline, Scopus, Web of Science, Cochrane and Google Scholar databases to identify studies on in vitro bonding of GIC (articles published until April 2021). The searching strategy was sound, and research risks were also shown. Realized analysis of in vitro studies evaluating bonding properties of GI restoratives to primary teeth suggests its good performance, especially for RM-GIC. Such analysis was performed to compare C-GIC and RM-GIC when applied in different substrates, such as enamel or dentin. The data were collected from publication details, characteristics of the material and samples, mean bond strength and failure mode. The article has shown that by measuring bond strength, the effectiveness of dental restoratives might be predicted. It has good scientific value to systematizing the methods and guides their advantages in tailoring the GIC. I appreciate showing these research results, but I think it would also be worth offering some images of in vitro results, mechanism of restorations with a different type of GIC.
Author response: We do appreciate the remark of the reviewer that the manuscript is a necessary addition to scientific literature in the field of dentistry. We also agree with you that adding images in meta-analysis would contribute to a popularity of the article. Unfortunately, we did not have authors` and/or publishers’ permission for such presentation.
Summing up, the presented review is essential for dentistry. The compilation of the results obtained for GIC materials in perspective will give the Authors and other researchers possibilities to produce the novel restorative materials. The manuscript has good scientific value but should be upgraded to attract the Readers' attention in such a form. As I read the manuscript, I also found some mandatory errors that the authors should correct before the publication. Some detailed suggestions are listed in the following points:
1. The conclusions in the article are too general. I suggest extending and re-editing the conclusions. I think The Authors should also highlight the main outlook of GIC applications here.
Author response: The conclusion has been edited as follows: “The meta-analysis of in vitro studies evaluating bonding properties of GI restorative materials to primary teeth suggests superior adhesion properties of RM-GIC to primary enamel and dentin, both immediately and after long-term storage. Finding an ideal restorative material for primary teeth is an ongoing quest, which could be speed up with further investigations of novel GIC. Development and standardization of test protocols, as well as extension of the storage period are pertinent in obtaining significant results. Continuous improvement of dental materials holds great promise for finding the ideal GI restorative for primary teeth.” (page 16)
2. Figures 2,3: The pictures should be upgraded. In such form, it has low quality of presentation and scientific value.
Author response: The presentation was improved and figures of better quality are presented in the revised Manuscript. We followed the Journal’s instructions for preparation of figures (minimum 1000 pixels width/height, resolution of 300 dpi or higher). Please also consider the original figures attached as a supplementary material. We presented the flow-chart and the forest plots for the meta-analysis in a usual manner for articles of this type. However, if the Reviewer and the Editor in Chief think that figures should be in color, we are more than willing to make such changes.
3. Sections should be more precisely named
Materials and Methods better to say: (Literature) Research Methodology
Results better to say (Literature) Research Results
Discussion of the Results
Author response: Although these modifications would make the sections more engaging to readers, we are not in a position to make changes, because we followed Journal Structure and Formatting recommendations (3.1. Overall Structure). We are more than willing to edit names of the sections if Editor in Chief approves such changes.
4. Materials and Methods section:
Line 4 – invitro / should be – in vitro
Author response: The suggested change has been done.
There are no more comments that I felt to comment on. In general, the article requires editorial refinement and proofreading. The issues presented in the article are important for researchers of primary teeth restoration processes and suitable for publication as a review article in the Journal Materials, but to increase their scientific value manuscript should be again analyzed by the Authors. I also want to point out that the Authors should format the manuscript according to the journal's and the MDPI publishing house guidelines. I recommend the paper for publication in Journal Materials after major revision.
Author response: We are thankful to the reviewer for the detailed guidance. We made necessary changes to the structure of the article according to the MDPI Style guide. We sincerely hope that the revised manuscript will be acceptable for publication in the MDPI.
Reviewer 2 Report
The article is well written and designed. I have following questions:
What was the conditioning (surface preparation) of the dentine surface before glassionomer application?
Explain why GMGI exhibited higher values than conventional GI. How chemical composition influences it?
Author Response
The article is well written and designed. I have following questions:
- What was the conditioning (surface preparation) of the dentine surface before glassionomer application?
Author response: The conditioning procedure has been explained in more detail as follows: “Dental tissues were conditioned in the following manner: 1) 20% polyacrilyc acid for 10 seconds [14,16], 2) 10% polyacrilyc acid for 20 seconds [17,18], 3) 10% polyacrilyc acid for 10 seconds [19,20], 4) 25% polyacrilyc acid for 10 seconds [15], or 5) Vitrebond primer- a mixture consisting of polycarboxylic acid and 2-hydroxyethylmethacrylate (HEMA) [15,19,20]”. (‘Results’ section, page 5, paragraph 1, lines 4-7)
- Explain why GMGI exhibited higher values than conventional GI. How chemical composition influences it?
Author response: A longer discussion on this topic has been included in the Discussion section: “The present analysis showed better bonding properties of RM-GIC to primary enamel and dentin compared to C-GIC, both immediately and after long-term storage. RM-GIC offer both micromechanical interlocking and chemical bonding to tooth substrate [30], and provide higher bond strength values and better long-term stability [15]. RM-GICs consist of two components: a conventional glass ionomer and a resin. Both components affect mechanical properties, setting mechanism, and adhesive strength. RM-GIC bond to tooth structure by two mechanisms: chemically, through ionic bonding of the carboxyl group to the calcium ions of the tooth substrate (typical for glass-ionomer component), and mechanically, by interlocking of the resinous component and conditioned tooth surface (polyacrylic conditioning provides slightly more retentive surface than not conditioned tooth structures) [31]. HEMA, as a component of RM-GIC, enhances monomer diffusion into the demineralized dentin matrix, and entanglement with its components. The addition of HEMA facilitates the formation of a “hybrid” layers. The “hybrid” layer is the formation of a transitional zone of resin-reinforced dentin, essential for achieving high bond strength [32].” (page 14, paragraph 1, lines 2-11)
To corroborate these statements, the following references have been added:
- Hamama, H.H.; Burrow, M.F.; Yiu, C. Effect of dentine conditioning on adhesion of resin‐modified glass ionomer adhesives. Aust. Dent. J. 2014, 59, 193-200.
- Nakabayashi, N.; Takarada, K. Effect of HEMA on bonding to dentin. Dent. Mater. 1992, 8, 125-130.
Reviewer 3 Report
This is a good review on a very important topic! Please let me ask if you could include more analyses regarding the results of the individual studies, e.g. how many percent of studies found RM-GIC to be better than C-GIC? how much % was the improvement on average? how many studies failed to find a significant difference?
Author Response
This is a good review on a very important topic! Please let me ask if you could include more analyses regarding the results of the individual studies, e.g. how many percent of studies found RM-GIC to be better than C-GIC? how much % was the improvement on average? how many studies failed to find a significant difference?
Author response: Interestingly, all studies revealed statistically significant difference in bond strength between RM-GIC and C-GIC, except one. We included the statement in the Results section as follows: “All authors reported significantly better performance of RM-GIC compared to C-GIC, except Pacifici et al. [16] who did not record significant difference between C-GIC and RM-GIC.” (‘Results’ section, page 10, paragraph 2, lines 2-4).